# Transcriptomic Changes of Murine Visceral Fat Exposed to Intermittent Hypoxia at Single Cell Resolution

**DOI:** 10.3390/ijms22010261

**Published:** 2020-12-29

**Authors:** Abdelnaby Khalyfa, Wesley Warren, Jorge Andrade, Christopher A. Bottoms, Edward S. Rice, Rene Cortese, Leila Kheirandish-Gozal, David Gozal

**Affiliations:** 1Department of Child Health, School of Medicine, University of Missouri, Columbia, MO 65211, USA; rene.cortese@health.missouri.edu (R.C.); gozall@health.missouri.edu (L.K.-G.); gozald@health.missouri.edu (D.G.); 2Department of Animal Sciences, University of Missouri, Columbia, MO 65211, USA; warrenwc@missouri.edu; 3Kite Pharma, Santa Monica, CA 90404, USA; andrade.jorge@gmail.com; 4Informatics Research Core Facility, Life Sciences Center, Missouri School, Columbia, MO 65211, USA; bottomsc@missouri.edu (C.A.B.); edsrice@gmail.com (E.S.R.)

**Keywords:** intermittent hypoxia, sleep apnea, OSA, single cell, snRNA-seq, bulk RNA-seq, deconvolution

## Abstract

Intermittent hypoxia (IH) is a hallmark of obstructive sleep apnea (OSA) and induces metabolic dysfunction manifesting as inflammation, increased lipolysis and insulin resistance in visceral white adipose tissues (vWAT). However, the cell types and their corresponding transcriptional pathways underlying these functional perturbations are unknown. Here, we applied single nucleus RNA sequencing (snRNA-seq) coupled with aggregate RNA-seq methods to evaluate the cellular heterogeneity in vWAT following IH exposures mimicking OSA. C57BL/6 male mice were exposed to IH and room air (RA) for 6 weeks, and nuclei from vWAT were isolated and processed for snRNA-seq followed by differential expressed gene (DEGs) analyses by cell type, along with gene ontology and canonical pathways enrichment tests of significance. IH induced significant transcriptional changes compared to RA across 14 different cell types identified in vWAT. We identified cell-specific signature markers, transcriptional networks, metabolic signaling pathways, and cellular subpopulation enrichment in vWAT. Globally, we also identify 298 common regulated genes across multiple cellular types that are associated with metabolic pathways. Deconvolution of cell types in vWAT using global RNA-seq revealed that distinct adipocytes appear to be differentially implicated in key aspects of metabolic dysfunction. Thus, the heterogeneity of vWAT and its response to IH at the cellular level provides important insights into the metabolic morbidity of OSA and may possibly translate into therapeutic targets.

## 1. Introduction

Sleep disordered breathing (SDB), more specifically obstructive sleep apnea (OSA), has been implicated as an important risk factor and potential cause of the metabolic syndrome [1,2,3]. The pathophysiology of the cardiometabolic complications in OSA is still incompletely understood; however, intermittent hypoxia (IH) as observed in OSA, and characterized by repetitive short cycles of oxyhemoglobin desaturation and reoxygenation, likely plays a pivotal role. Furthermore, emerging evidence of a relationship between OSA and metabolic perturbations, and in particular with alterations in glucose metabolism such as insulin resistance (IR) and type 2 diabetes (T2D) has been reproducibly reported across a multitude of studies [4,5,6,7,8,9,10,11,12,13,14,15], including in healthy volunteers [16,17,18]. To elucidate the potential mechanisms implicated in such metabolic derangements, murine models have been developed that traditionally include one of the major perturbations of OSA, namely IH [1,19,20]. 

The visceral white adipose tissue (vWAT) has emerged as a highly active endocrine organ, [19]. It plays a key role in metabolism, which is mediated predominantly through the secretion of multiple hormones, cytokines, chemokines and other proteins, collectively referred to as adipokines [21,22]. Evidence from the obesity literature has implicated hypoxia in vWAT as a major driver of metabolic dysfunction and insulin resistance (IR) [23]. Thus, the level of hypoxia in WAT could be further aggravated and its patterning altered when obese patients also suffer from OSA [19]. Adipose tissue is heterogeneous and comprised of multiple cell types including adipocytes, preadipocytes, endothelial cells, stromal cells, and several immune cell subtypes [24,25]. Adipocytes are the major functional cell type in adipose tissues, making up 20% of total cells. However, about 80% of cells in adipose tissues include a complex mixture of stromal cells such as fibroblasts, vascular cells, adipocyte stem cells and immune cells [26,27,28,29]. The composition of stromal cells varies across fat depots, which likely reflects tissue specialization and differences in energy storage, vascularization, innervation, and metabolism [27,30]. By understanding better the cellularity of adipose tissue, its variability in the population, and the individual and collective roles played by these cellular subsets in both health and disease, we should gain improved insights into the central role of vWAT in metabolic dysfunction induced by IH mimicking OSA. 

Recently, technologies have been developed that allow simultaneous single-nucleus mRNA sequencing (snRNA-seq) of thousands of cells originating from adipose tissues, and thereby provide information that is largely unbiased and far more comprehensive than approaches based on bulk RNA transcriptomic analysis [31,32,33]. snRNA-seq has provided unique information and insights into the developmental lineages underlying multicellular organisms [34,35,36], has facilitated the discovery of novel cell types, and revealed the intricacies of multicellular regulatory networks along with interactive gene networks in both health and disease [37,38,39,40,41]. In addition, these technologies can effectively isolate the gene regulatory signals from rare cell populations to infer potentially useful determinants of specific phenotypes, and also serve as disease markers, and can be employed in studies of cell lineage and regulation of differentiation [42,43]. Two recent studies have reported on the use of snRNA-seq in adipocytes, either in isolated brown adipocytes or from inguinal white adipocyte nuclei [44,45]. Furthermore, snRNA-seq was employed in adipose tissues from murine models and in human subjects [45,46,47,48,49,50].

To improve our understanding of the perturbations occurring in chronic IH emulating OSA that lead to insulin resistance, we combined snRNA-seq with bulk vWAT RNA-seq in a mouse model. We hypothesized that the cell heterogeneity of vWAT in IH would differ from the control conditions of normoxia, and that such changes within specific cell types may provide important information that relates to the emergence of insulin resistance in vWAT. 

## 2. Results

### 2.1. snRNA-seq Analysis

We used vWAT for snRNA-seq and bulk RNA-seq as shown in Figure 1A, after ascertaining sample quality that included nuclei purity imaging and cDNA library size distribution (Appendix A). A featured plot was applied to identify nuclei outliers, and this plot showed the number of gene counts 20–4500 (*Y*-axis) and the number of RNA reads 0−50,000 (*X*-axis) (Appendix A). The percentage of measured gene expression in each nucleus was attributed to mitochondrial genes (Appendix A) and the number of unique genes that was found for each nucleus was within expected values for each cell [51] (Appendix A). To identify low-quality cells and doublets, we looked at the distribution of the percentage of mitochondrial genes expressed, unique molecular identifier (UMI) in each cell, and the number of genes expressed in each cell. 

After quality control scrutiny and exclusion, ten cell types (0–9) were identified in RA compared to 13 cell types (0–12) in IH (Figure 1B–C), and the heatmap for the top five genes for each of the cell types is shown in Figure 1D–E. The cell types identified in vWAT of mice exposed to IH were clearly distinct from mice exposed to RA. The annotation of differentially expressed genes (DEGs) for each cell types (Table 1) showed that unsurprisingly adipocytes were the most prevalent cell type in both IH and RA groups (Figure 1B–C). In RA conditions, the following cell types were identified: neurons-1, smooth muscle cells, adipocytes, fibroblasts, germ cells, macrophages-1, macrophages-2, endothelial cells, neurons-2, and mesothelial cells. In IH conditions, the following cell types were identified: germ cells, podocytes, and retinal ganglion cells, B cells, oligodendrocyte progenitor cells, macrophages, adipocytes-1, endothelial cells, neurons, adipocytes-2, adipocytes-3, adipocytes-4, and pericytes. 

To gain broader insight into the gene expression changes occurring in vWAT of mice exposed to IH and RA within each cell type classification, we identified a total of 4810 DEGs across all cell types. Furthermore, to identify genes that are enriched in each specific cell type, the mean expression levels of each gene were calculated across all cells, then each gene from each cell was compared to the median expression of the same gene from cells in all other cells types. The top 5 ranking markers revealed distinct signatures of vWAT as shown in the heatmap among cell types (Figure 2A and Table 2).

### 2.2. vWAT Cell Type Composition Following IH and RA Exposures

Following cluster identification across all samples, cellular putative identities were integrated and resulted into a clear delineation of 14 cell types. These analyses defined the baseline adipose tissue matrix from largest to smallest as luteal cells (12.3%), endothelial cells (12.1%), enterocytes (11.3%), oligodendrocytes (10.2%), T memory cells, B cells (10.1%), germ cells (9.6%), podocytes (7.3%), adipocytes (7.3%), enterocytes (5.3%), smooth muscle cells (3.5%), mesothelial cells (2.8%), and pericytes (2.8%) (Figure 2B). We found that adipocytes and endothelial cells constitute the largest proportion (19.4%), while mesothelial and pericytes cells account for only a small percentage (2.8%) of cells. These findings suggest that tissue cellularity and metabolic responsiveness may be intimately dependent on the cross talk between the cells making up vWAT.

### 2.3. Gene Ontology and KEGG Enrichment Analyses of snRNA-seq Genes

To infer the biological properties of each of the cell types within vWAT in the context of IH exposures, we performed Gene Ontology (GO) and Kyoto Encyclopedia of Genes and Genomes (KEGG) enrichment assessments (Appendix A). GO analysis was performed on three different aspects including biological process (BP), cellular component (CC) and molecular function (MF) for all DEGs in all cell types (Appendix A). We also performed KEGG pathway enrichment analysis for DEGs, and the significant KEGG pathways and their genes are shown in (Appendix A). For the sake of brevity, we highlight here some of the salient pathways with potential biological significance in IH, such as: Rap1 signaling pathway (KEGG:04015), *p*-value 2.21 × 10^−20^; focal adhesion (KEGG:04510), *p*-value 5.04 × 10^−20^; platelet activation (KEGG:04611), *p*-value, 9.64 × 10^−16^; phospholipase D signaling pathway (KEGG:04072), *p*-value 5.58 × 10^−14^; regulation of actin cytoskeleton (KEGG:04810), *p*-value 1.86 × 10^−12^, pathways in cancer (KEGG:05200), 2.08 × 10^−12^; and PI3K-Akt signaling pathway (KEGG:04151), *p*-value 3.16 × 10^−12^ (Appendix A). 

We also performed GO and KEGG for each of the cell types, and as would be anticipated different cell types exhibited functions that were distinct from each other following IH exposure. However, some pathways were shared across multiple subpopulations (Appendix A). Thus, there appeared to be little functional overlap between the cell clusters, thereby indicating that each cell subpopulation had a unique array of functions. For example, cell types (clusters 0–3, smooth muscle cells; enterocytes; macrophages; and T memory cells, B cells) exhibited enrichment in genes related to ECM-receptor interactions, focal adhesion, regulation of actin cytoskeleton, Ras signaling pathway. However, macrophages in this group also manifested heightened expression of pathways such as PI3K-Akt signaling, Rap1 signaling, Ras signaling, ECM-receptor interaction, MAPK signaling, TGF-beta signaling, and EGFR tyrosine kinase inhibitor resistance. Cluster 3 (T memory cells) in IH revealed recruitment of pathways underlying natural killer cell-mediated cytotoxicity, phosphatidylinositol signaling system, cancer, leukocyte trans-endothelial migration, T cell receptor signaling, VEGF signaling pathway, MAPK signaling pathway, sphingolipid metabolism and signaling, calcium signaling pathway, inositol phosphate metabolism, non-small cell lung cancer, and cAMP signaling. In IH-exposed mice, clusters 4 and 8 (adipocytes) expressed gene pathways specifically related to metabolic pathways, AMPK signaling, regulation of lipolysis in adipocytes, insulin resistance and signaling, glucagon signaling, PPAR signaling, adipocytokine signaling, cAMP signaling, beta-alanine metabolism, fatty acid biosynthesis and metabolism, pyruvate metabolism, PI3K-Akt signaling, sphingolipid signaling, adrenergic signaling, and HIF-1 signaling. When exploring specific genes within specific relevant pathways related to metabolic deregulation induced by IH in adipocytes clusters (4 and 8), a high enrichment of genes involved in the regulation of lipolysis (mmu04923) such as ADRB3, PLIN1, ADCY5, PDE3B, TSHR, LIPE, and insulin signaling pathway (mmu04910) such as PRKAR2B, SORBS1, ACACA, PDE3B, LIPE, PCK1. Similarly, in cluster 2 (macrophages) genes associated with pI3K-Akt signaling pathway (mmu04151), such as AKT3, COIIA2, CREB5, CSF1, EGFR, FIGF, FN1, and ITGA11 were highly regulated in IH. Similar enhancements in functionally relevant pathways were also identified in other clusters. These data indicated that IH exposures targeted distinct, highly metabolically active macrophage and adipocyte populations in vWAT. A comprehensive list of the highly regulated genes for each pathway is provided in Appendix A.

To find cell-type specific gene expression changes in vWAT of mice exposed to IH compared to RA, genes from each cell type were compared. We found genes associated with adipogenic differentiation, including master regulators Pparg (cluster 4) and Cebpa (cluster 10), several genes underlying adipogenesis, such as Klf2 (cluster 7, 10, 11), Ebf1 (clusters 2, 5, 6, 9), and Ebf2 (clusters 2, 5) and common adipocyte markers, including Pparg (cluster 4, Cebpa (cluster 10), Adipoq (cluster 6, 9, 10), and Lep (clusters 6, 8, 9, 10). Examples of genes that correlated positively with adipocytes include Pparg, and genes involved in lipid synthesis, such as Fabp4, Acsl1, and Lpin1, as well as those involved in mitochondrial electron transport, i.e., Atp5b. Indeed, the KEGG and GO analyses show that vWAT in IH was enriched in many pathways related to metabolic processes, especially of lipids, such as oxidative phosphorylation, glycerolipid metabolism, and arachidonic acid metabolism. Furthermore, when examining adipocyte subtypes (1 and 2) and we found that DEGs in these cells play an important role in pathways related to metabolic processes such as AMPK signaling pathway and regulation of lipolysis in adipocytes. In addition, despite differentiation of transcriptomes, i.e., separate clustering (cell types), for both adipocyte subtypes, we find overlapping enrichment of genes involved in insulin signaling pathway such as PRKAG2, HK2, ACACA. Interestingly, we also identified several genes involved in adipogenesis such as Fabp4 among nonadipocyte cell types (endothelial cells, enterocytes, and oligodendrocytes), Fasn (enterocytes), Pparg (adipocytes-1, luteal cells, endothelial cells, adipocytes-2, and germ cells) and Lpl (luteal cells, adipocytes 2, and enterocytes). In addition, several genes that are involved in lipid mobilization, such as Adrb3, Lipe, and Pnpla2, and genes associated with adipocytes, such as Nrip1, Lpl, Zbtb20, and pan-adipocyte genes such as Cd36, Fabp4, and Aqp1 were also highly expressed, suggesting that these cells may be involved in lipid mobilization and metabolic dysfunction induced by IH in vWAT. 

### 2.4. Transcription Factors Enrichment by Cell Type

A catalog of transcription factors involved in coordinating metabolic response to IH and RA is another level of information necessary for disease interpretation. To reveal key transcription factors associated with DEGs in RA compared to IH of each cell type, we queried the gene regulatory networks TRANSFAC (https://biit.cs.ut.ee/gprofiler/gost) database. We first calculated the number of TFs per cell type, and plotted the number of genes and TFs that were detected in individual and all combined cell types (Figure 3A). As shown in Figure 3A, endothelial cells and oligodendrocytes have the highest number of TFs compared to other cell types. Across all 618 TFs associated with all DEGs, we found 125 TFs for both adipocyte subtypes in both clusters (Figure 3B). Next, we used the DEGs that were identified in adipocyte cells (clusters 4 and 8), and inserted them into the WEB-based GEne SeT AnaLysis Toolkit, WebGestalt, [52] to identify candidate genes for a disease or set of abnormal phenotypes. The data showed the frequency of each KEGG (Figure 3C) and phenotypes (Figure 3D) in the feature subset and displayed the ratio of the number of each KEGG term to the scale of the number of diseases. In adipocyte cell clusters, several phenotype-based predictions of disease-related genes are associated with genes in adipocytes, including regulation of lipolysis, adipocytokine signaling, and insulin resistance (Figure 3D). 

### 2.5. Adipocytes and Adipogenesis Genes

Of a set of 25 known adipocyte and adipogenesis genes, we found that 20 genes are associated with different clusters as shown in Figure 4A. We also show individual gene expression comparisons for several genes (Figure 4B). We found that LpI, Fabp4, Insr, and Pparg were ubiquitously expressed in most cells, while Adipoq and Lep were expressed only in adipocytes subtype. 

To further interrogate the cell types and functions within cell types, we used GO and KEGG pathways. Several KEGG pathways were identified (*p* < 0.05): signaling pathway; PPAR signaling pathway; nonalcoholic fatty liver disease (NAFLD); insulin resistance; insulin signaling pathway; regulation of lipolysis in adipocytes; adipocytokine signaling pathway; fatty acid biosynthesis; pathways in cancer; and transcriptional mis-regulation in cancer. The top four significant GO pathways for CC included extracellular space, RNA polymerase II transcription factor complex, membrane-bounded organelle, lipid droplet, and for BP: regulation of lipid localization, fatty acid metabolic process, response to organic substance, and response to endogenous stimulus. For MF pathways, signaling receptor binding, hormone receptor binding, monocarboxylic acid binding, and carboxylic acid binding emerged (Appendix A). The DEGs of the selected genes were further analyzed using pathway functional analysis, and found that these genes are involved in various highly significant KEGG pathways (*p*-value 9.80 × 10^−10^ to low 6.0 × 10^−12^). Among those highly significant pathways, AMPK signaling pathway, PPAR signaling pathway, nonalcoholic fatty liver disease (NAFLD), insulin resistance, insulin signaling pathway, regulation of lipolysis in adipocytes, adipocytokine signaling pathway, and fatty acid biosynthesis are noteworthy. Furthermore, we constructed gene networks of adipocyte-selected genes using the STRING tool (https://string-db.org/) as shown in Figure 4C. We found Cebpa, Fabp4, lep, Adipoq and pparg were connected as the main network, while klf3 and Retnl3 were not connected to the same network suggesting differential roles. 

### 2.6. RNA-seq Analysis of vWAT in RA and IH Mice

To identify and characterize the global differentially expressed genes in vWAT of mice exposed to IH, we used bulk RNA-seq approaches. We identified a total of 617 DEGs using 1.5-fold changes and *Q*-value < 0.05 criteria. Of the 617 genes, there were 299 upregulated and 318 downregulated genes between IH vs. RA conditions. Heatmap for the expression values of the two groups revealed accurate group assignment, indicating the presence of uniquely and significantly different expression patterns for the DEGs (Figure 5A). To evaluate the biological roles of the DEGs, GO including MF, BP, and CC; and KEGG pathway enrichment analyses were applied (Figure 5; Appendix A). The GO and KEGG for the upregulated DEGs is shown in Figure 5B, and for the downregulated DEG is shown in Figure 5C. In addition, the GO and KEGG for all DEGs are shown in Appendix A. These DEGs are involved in various KEGG pathways, including metabolic pathways, ECM-receptor interaction, nonalcoholic fatty liver disease (NAFLD), focal adhesion, PI3K-Akt signaling pathway, regulation of lipolysis in adipocytes, AMPK signaling pathway, insulin resistance, and sphingolipid metabolism. The list of genes involved in each pathway is shown in Appendix A. 

We used the top 30 DEGs (15 upregulated and 15 downregulated) to construct gene-gene interactions using the STRING software. This analysis showed 426 nodes with 1235 edges and average node degree of 7.1, average local clustering coefficient of 0.363, and protein−protein interaction (PPI) enrichment *p* < 1.58 × 10^−12^ (Figure 5D). In this network, leptin was located at the center of the networks and connected with many other metabolic genes.

### 2.7. Transcription Factor Analysis in Bulk RNA-Seq

Next, we also used the TRANSFAC software to identify transcription factors (TFs) for the DEGs in the bulk RNA-seq data (Figure 6A). First, we used the 617 DEGs, then only upregulated genes, and thirdly, we used only downregulated genes (Figure 6A). We found 292 TFs in all DEGs, 221 in upregulated and 15 in downregulated genes (Figure 6A). In all DEGs, we found that the TFs (SP, E2F, and Ap2) were more abundant compared to other TFs (Figure 6B). In upregulated DEGs, we found TFs (SP, E2F, AP2, and EGR) as being more abundant (Figure 6C), while in downregulated DEGs, we found that TFs (SP1 and CTCF) were most abundant (Figure 6D). 

### 2.8. Comparison of DEGs in snRNA-seq and Bulk RNA-seq

DEGs identified in snRNA-seq and in bulk RNA-seq were compared using a Venn diagram (Figure 7A). Out of the 617 DEGs in bulk RNA-seq and 4810 DEGs in snRNA-seq, 319 genes were uniquely expressed in bulk RNA-seq, while 2659 were exclusively expressed in snRNA-seq, and 298 genes were common DEGs between the two approaches. Next, we identified the KEGG pathways in these three groups (Figure 7A), and found that insulin resistance and PI3K-AKT pathways were highly enriched. We further used gene network analyses to illustrate the molecular regulator and connections for each of these pathways, insulin resistance (Figure 7B), and PI3K-AKT (Figure 7C). Interestingly, in PI3K-AK, their three different networks were not connected to one another indicating that those genes may act individually or can be connected under different conditions. 

### 2.9. Deconvolution of Bulk RNA-seq

We estimated cell-type proportion utilizing CIBERSORTx using the bulk RNA-seq gene expression data to construct a tissue-specific signature matrix. The cell proportion estimates produced by CIBERSORTx are shown in Figure 8. Th adipocyte proportions were 0.51 for RA and 0.73 for IH (*p* < 0.03). Proportions of the other estimated cell types ranged from 0.173 for RA and 0.216 for IH for macrophage (M0; *p* < 0.04), 0.188 for RA and 0.242 for IH for monocytes (*p* < 0.005), while B-cell displayed 0.072 for RA and 0.186 for IH (*p* < 0.007). *n* = 3/condition.

### 2.10. Verification of Bulk RNA-seq and snRNA-seq Using qRT-PCR 

To validate the DEGs of bulk RNA-seq in vWAT, qRT-PCR was performed. The genes used for validation were identified from the overlap between bulk RNA-seq and snRNA-seq that were involved in insulin resistance as shown in the Venn diagram (Figure 7A). We selected six genes that displayed underlying insulin resistance of which three genes were upregulated and three were downregulated in IH vs. RA conditions (Figure 9). For the upregulated genes, namely Slc27a1, PIK3CB, and Pik3r1, findings were highly significant (*p*-value < 0.001), and for downregulated genes, confirmatory findings were also corroborated (*p*-value < 0.05), with estimates similar and concordant with the same DEGs from bulk RAN-seq data. 

## 3. Discussion

Adipose tissue contains different cell types that play a prominent role in tissue homeostasis and are susceptible to major perturbations leading to insulin resistance and metabolic dysfunction following IH exposure mimicking sleep apnea. SnRNA-seq revealed an unexpectedly broad repertoire of cells that appear to mediate complex functions and are altered by chronic IH during sleep. Indeed, using snRNA-seq, bulk RNA-seq, and deconvolution of bulk RNA-seq to investigate the cell-type composition of tissue and heterogeneity, we clearly identified 14 cell types and among those, 3 specific subpopulations, namely macrophages, adipocytes, and endothelial cells appeared to participate in a distinct metabolic pathway. In each of the cell types, we identified numerous genes, which aligned with known GO terms, KEGG enrichment for each cell type, and identified candidate transcription factors for each cell type. In addition, the deconvolution of bulk RNA-seq allowed us to identify several cell types that were related to adipocytes, B-cell macrophages, and T cells. Furthermore, we found that the overlapping DEGs between snRNA-seq and bulk RNA-seq that enlighten the unique advantages of snRNA-seq relative to bulk RNA-seq approaches while revealing robust concordance on metabolic function and insulin resistance pathways. Our findings provide novel cell heterogeneity characterization of vWAT in mice exposed to IH, and propose that interrogation of complex tissues at the level of individual cells is essential to gain increased understanding of alterations in organ homeostasis and the pathological changes induced by OSA.

The pathophysiology of metabolic complications among OSA patients remains unclear; however, the repetitive short cycles of desaturation and reoxygenation, i.e., IH, clearly play a pivotal role [53]. Hypoxia causes vasoconstriction in the systemic and pulmonary circulation due to sympathetic nervous system activation, and recurrent hypoxia triggers systemic inflammation; alteration in transmural, intrathoracic, and cardiac pressure [8,54]. Furthermore, upper airway anatomy and collapsibility remain a fundamentally important pathophysiological factor. However nonanatomical factors, such as impaired muscle responsiveness, low arousal threshold, high loop gain, rostral fluid shifts, lung volume, additionally play a variable role [55,56]. Recent focused studies in murine models and in humans indicate that IH mediates some of its detrimental effects through adipose tissue inflammation and dysfunction [1,20,28,57,58,59,60,61,62,63,64,65]. However, adipose tissues are comprised of a heterogeneous collection of cell types [24], such that traditional microarray and bulk RNA-seq technologies, which provide the average gene expression level of all cells in each tissue, may not necessarily reflect the unique transcriptomic changes that occur in each individual cell and as such may provide misleading information as to the roles played by critical cellular subsets in the emergence of metabolic dysfunction. In contrast, snRNA-seq enables the quantification of the gene expression distribution across cells and in individual cells [66,67]. snRNA-seq has been used to study different organs [68,69,70] including various regions of the brain [71,72], immune cells [73], and hematopoiesis [74], as well as interrogate and classify aortic macrophage heterogeneity at the single-cell level in atherosclerosis [75]. It has been reported that adipocytes are raised from distinct developmental lineages and exhibit depot-specific differences in adipokine release, lipolysis, and inflammation [76,77]. Recently, two studies have reported on distinct inguinal white adipose tissue adipocytes, i.e., brown adipocytes or white adipocytes using scRNA-seq or snRNA-seq, respectively [44,45]. Here, using snRNA-seq, we identified adipocytes in two clusters (4 and 8) and macrophages in cluster 2. The cells in cluster 10 were identified as enterocytes, which are part of the immune system and contain the intracellular organelles, such as mitochondria, lysosomes, and endoplasmic reticulum, common to all cells. These enterocytes support normal cellular functions, and enterocytes exist in close association with tissue macrophages, whose activation during inflammatory processes leads to the release of nitric oxide (NO) [78]. Endothelial cells (ECs) are well known to play an important role in adipose tissue inflammation [79], and scRNA-seq approaches used to study ECs heterogeneity confirm this role [80,81,82]. Cluster 13 was identified as pericytes, and these cells are associated with endothelial cells and to stabilize nascent vascular networks [83].

scRNA-seq was also used in the study of mouse adipose tissue progenitor cells [50,84], and subcutaneous adipose tissues [45]. Mesothelial-like cells were identified in scRNA-seq [46,50]. These mesothelial cells are within thin membranes containing a layer of epithelial-like cells surrounding some internal organs, including visceral vWAT depots which act as a source of adipocyte progenitors during development [85,86,87]. Furthermore, mesothelial cells are highly responsive to inflammatory signals and secrete high levels of IL-6 and IL-8 following stimulation, suggesting a potential role for mesothelial-derived adipocytes in the inflammatory response in visceral fat [88]. scRNA-seq was used to characterize immune cell populations during vWAT remodeling, either following ADRB3 stimulation or during diet-induced obesity [49,89]. ADRB3-induced remodeling in vWAT is associated with the rapid accumulation of adipose tissue macrophages that are involved in phagocytosis and lipid clearance [90,91]. Recruitment of macrophages exhibit heterogeneity in phenotypes depending on the type and duration of the stimulus [91,92]. In addition, macrophages support angiogenesis in other organs, promoting blood vessel formation or expansion, providing survival and migratory cues to endothelial cells, and facilitating bridging of vascular growth. In our study, we identified PDGFRA in podocyte cells, and this marker was identified in cells undergoing active adipogenesis [50]. Furthermore, we identified retinol-binding protein 4 (RBP4) in germ cells and enterocytes cells as being upregulated in IH compared to RA. RBP4 is produced by visceral adipocytes and other tissues, and can activate and promote adipose tissue inflammation, which contributes to insulin resistance [93,94]. Furthermore, FN1 gene was found in macrophages, T memory cells, and B cells and is considered as an extracellular matrix glycoprotein that participates in cell differentiation, growth, and migration, and is involved in tissue remodeling and wound healing [95].

Smooth muscle cells (SMCs) play an essential role in maintaining the structural and functional integrity of blood vessels, and thus are a critical element for blood vessel construction via tissue engineering approaches. We found that SMCs share similarity with gene markers such as Adamtsl1, Rbfox1, Il1rapl1, and Mast4. One of the KEGG pathways that was identified in SMCs was extracellular matrix (ECM)-receptor interaction, and this ECM in vWAT is associated with insulin resistance, diabetes and obesity, and is crucial for the expansion of the vWAT to allow necessary and proper structural changes [96,97]. Furthermore, in our bulk RNA-seq data, we found that ECM is the top of KEGG pathway with highly significant and fold changes. We also found that ECM in other cell types indicating that ECM may play an important role in metabolic dysfunction. 

Bulk RNA-seq profiling is widely used for biomarker discovery, genetics of gene expression studies, and differential expression analysis [98,99], and can also be used for identifying cell-type heterogeneity which can play an important role in understanding the interrelationship between cell-type proportions and complex diseases [100]. Furthermore, due to the complexity of cell-type composition deconvolution, algorithms can be applied to computationally estimate cell-type proportions from gene expression data of tissues or blood [101]. Several studies have utilized cell-type specific gene expression profiles derived from scRNA-seq for cell composition deconvolution and among those methods CIBERSORT is highly utilized [102,103,104,105]. Accordingly, we used CIBERSORTx to estimate the fraction of immune systems and adipocytes in RA and IH, and established regulatory coexpression patterns based on correlation analyses between immune cells, genes, and signaling pathways. We found that adipocytes were the highly significant cells in IH compared to RA. The majority of the data that was used in deconvolution was related to referenced profiles of gene expression from other studies which did not take into account the fact that the reference expression profiles were often disturbed by microenvironment or developmental effects or were simply obtained under different conditions or with different technologies or platforms. In our studies, we used vWAT derived from the same mice for both snRNA-seq and bulk RNA-seq. Therefore, current findings indicate that cell-type deconvolution algorithms can be used to make inferences about cell-type composition at the macrophage and adipocyte level and therefore acquire improved insights into the perturbations elicited by IH in this setting. Furthermore, another method has been used to provide cell composition in different tissue using immunohistochemistry, however, immunohistochemistry is useful for studying cell type composition and spatial organization, but is limited by its use of only preselected marker genes, and cannot be used for further genome-wide expression analysis [106]. It has been reported that some discrepancies between cell composition revealed by scRNA and deconvolution of bulk RNA-seq. For example, several methods have been used for a decomposing bulk expression such as CIBERSORTx, originally designed for microarray data, that utilizes a reference generated from purified cell populations. A major limitation of this approach is the reliance on sorting cells to estimate a reference gene expression panel [107]. Most bulk RNA-seq datasets do not have corresponding snRNA-seq data in the same set of samples. Another limitation is that the identification of cell-type methods, because of the differences in the capture of mRNA for bulk RNA-seq and the chemistry used for snRNA-seq technologies, may present an issue for decomposition models that assume a directly proportional relationship between the snRNA-seq based reference and observed bulk mixture [107,108]. Recently, Bisque was suggested for better deconvolution as it is a highly efficient tool for measuring cellular heterogeneity in bulk expression through robust integration of single-cell information, accounting for biases introduced in the snRNA-seq methods [109].

Since transcriptional regulation of gene expression plays a critical role in many cellular processes [110,111,112], and since identification of functional transcription factors is essential for understanding gene regulatory mechanisms, we sought to explore putative TFs potentially involved in our gene expression findings. Here, we are identifying candidate TFs in each cell type and further studies will be followed on their function and epigenetic changes. In addition, we should emphasize that one TF can regulate different genes in different cell types [113]. In our current study, we identified all known TFs in both snRNA-seq, either at the single cell level or in the bulk RNA-seq data. We found that the most abundant Ts in each cell type were specificity protein (SP). E2F transcription factors which regulate adipocyte differentiation [114] were also abundantly represented. The zinc finger transcription factor early growth response-1 (EGR1) is expressed in adult adipose tissues, where its overexpression has been linked to obesity in both humans and mouse models [115]. Consistently, EGR1 inhibits lipolysis and promotes fat accumulation in cultured adipocytes by directly repressing the transcription of the adipose triglyceride lipase gene [116].

The strength of our study was for the first time to show the cell types of eWAT in mice exposed to IH using both bulk RNA-seq and snRNA-seq technologies from the same tissues, and comparing and identifying DEGs in both technologies as well as identifying candidate TFs for each cell type. However, there were some limitations, such as animal numbers and quantity of adipose tissue for further validation of targets using qRT-PCR.

We also compared DEGs in snRNA-seq and bulk RNA-seq from vWAT of mice exposed to IH, and found a considerable overlap in gene expression between both approaches in the same tissues from the same mice. These overlapping genes were mainly involved in metabolic dysfunction and insulin resistance pathways, suggesting a transcriptional manifestation of tissue-specificity in adipose tissues that was established during IH exposure.

## 4. Materials and Methods

### 4.1. Animals

Non-obese C57BL/6 male mice (Jackson Laboratory, Bar Harbor, ME, USA) were housed 5/cage at 24 ± 1 °C for 12-h light/dark cycle with unrestricted ad libitum access to standard chow and water. 

### 4.2. Intermittent Hypoxia (IH)

IH was performed according to our standard published protocols [20,62,117,118]. Mice were exposed to IH or normoxia (room air, RA) for 6 weeks (5 mice/cage) using standard environmental chambers operated by a commercial system and software (Oxycycler A44XO, BioSpherix, Parish, NY, USA). The pattern of IH consisted of alternating cycles of 90 s (6.5% FiO_2_ followed by 21% FiO_2_) for 12 h per day (7:00 a.m. to 7:00 p.m.), while RA mice were exposed to 21% FiO_2_ throughout. The oxyhemoglobin saturation at the end of the hypoxic period was reaching 68–75%, mimicking values commonly experienced by OSA patients. 

### 4.3. Epidydimal Visceral White Adipose Tissues (vWAT)

vWAT were dissected from mice exposed to IH or RA for 6 weeks and tissues were immediately stored at −80 °C until processing.

### 4.4. Single Nuclei RNA-Sequencing (snRNA-seq)

Single nuclei were prepared from vWAT samples (100 mg of tissue each), and nuclei were isolated using a Dounce homogenizer (VWAR, Batavia, IL, USA) in RNase-free lysis buffer (10mM Tris-HCl, 10mM NaCl, 3mM MgCl2, 0.1% NP40), followed by lysis at 4 °C for 5 min. The samples were centrifuged at 500× *g* for 7 min, the supernatant was removed, and the cell pellet was resuspended in 1 mL resuspension buffer (1xPBS with 1% BSA and 0.2U/ul RNase Inhibitor). The cells were filtered through a 40 µm filter (pluriSelect Life Science, El Cajon, CA, USA) and centrifuged at 500× *g* for 7 min at 4 °C. Nuclei counts were performed using a Countess II Automated Cell Counter (Life Technologies, Carlsbad, CA, USA) to determine the volume suspension to be used in the 10x Chromium Single Cell Library protocol. A dilution of collected nuclei were also stained with DAPI (4′,6-diamidino-2-phenylindole, 10 ug/mL) to perform differential interference phase microscopy (DIC) visualization. 

We targeted ~4000 nuclei for each sample, which is consistent with prior studies [119,120], *n* = 2/condition. snRNA-seq libraries were constructed using the Chromium Single Cell 3′ library with gel beads in emulsion (GEMs), v3.1 (10× Genomics, Pleasanton, CA, USA), according to the manufacturer’s protocol. Nuclei suspensions, reverse transcription master mix, and partitioning oil were loaded on a single-cell chip, and then run on the Chromium Controller. Libraries were sequenced on an Ilumina NextSeq (Illumina, San Diego, CA, USA) with target reads per cell of 25,000. cDNA libraries were sequenced using NextSeq 600 (Illumina, San Diego, CA, USA).

### 4.5. snRNA-seq Bioinformatics Analysis

Reads from single nuclei were initially aligned against the mouse reference (GRCm38) and quantified using Cell Ranger v3.0.2, and thereafter all analyses were performed with various R modules of Seurat v3.0 [121]. The quality of raw reads was assessed by FastQC, followed by trimming of adapters using TrimGalore, and reads that passed the raw data QC were used for mapping using STAR [122]. A transcript compatibility count per sample was then calculated. Nuclei with high mitochondrial expressed genes (>20%) were removed, as high mitochondrial expression often indicates cells undergoing apoptosis. High outliers for the UMI count per nuclei were also removed as possibly multiple nuclei captured in a single GEM. Nuclei were filtered based on a minimum number of 200 and a maximum number of 5000 expressed genes per nuclei.

Aggregated data were subjected to Louvain cluster analysis for cell type identification using the R package Seurat (v3.0). Each cluster was labeled as the most frequent cell type across all its marker genes, with each label associated with a gene weighted by the average log fold change. Dimensionality reduction was then performed using T-distributed stochastic neighbor embedding (t-SNE) and uniform manifold approximation and projection (UMAP) tools. The detection of differentially expressed genes (DEGs) was performed using FindMarkers function in Seurat with *p*-value adjustment using the Bonferroni correction with *p*-value < 0.05. DEGs in each cluster were defined by log2 fold change of ≥ 2 compared with expression in all other clusters. Fold changes of DEGs were calculated using Seurat log2 (avg. of RA/avg. of IH). The selection of the top 5 DEGs for each cluster was based on the lowest adjusted *p*-value in order to identify cell types for each cluster in both RA and IH conditions. To partition cells into clusters, we used the smart local moving (SLM) algorithm for modularity-based clustering, and for each data set, we computed a cell–cell distance matrix constructed on selected aligned canonical correlation vectors [123]. We constructed a shared nearest neighbor (SNN) graph based on this distance matrix to use as input to the SLM algorithm, implemented through the FindClusters function in Seurat. To visualize the resulting clusters in two dimensions, we used Barnes–Hut implementation of the t-distributed stochastic neighbor embedding (t-SNE) algorithm) [124]. We used the mouse PanglaoDB39 [125] database to identify cell type for the top 5 DEG from each cluster. 

### 4.6. RNA-seq Analysis of vWAT

Total RNAs were isolated from vWAT using RNeasy Lipid Tissue Mini Kit with DNase treatment (Qiagen, Valencia, CA, USA) as described [126]. The RNA quality and integrity were assessed using the Eukaryote Total RNA Nano 6000 LabChip assay (Agilent Technologies, Santa Clara, CA, USA) on an Agilent 2100 Bioanalyzer (Thermo Fisher Scientific, Waltham, MA, USA). RNA samples were quantified by measuring A26 0 nm on a UV−vis spectrophotometer (ND-1000, NanoDrop Technologies, Wilmington, DE, USA). The RNA quality was 1.9–2.0 using NanoDrop, and the RNA integrity was 9.8–9.9 based on the RNA Integrity Number (RIN). The RIN is a software tool that scans the peaks of RNA electropherograms for RNA intactness (Appendix A). Poly-A enriched mRNASeq libraries were prepared following Illumina’s TruSeq Stranded mRNA LT library preparation protocol (Illumina Inc., San Diego, CA, USA) using 1 μg of total RNA. The cDNA sequencing libraries were generated from poly-A selected RNA using a TrueSeq library preparation kit (Illumina, Inc., San Diego, CA, USA). All sequencing was performed on an Illumina HiSeq 2500 (Illumina, Inc., San Diego, CA, USA). 

The raw RNA-seq data were analyzed by using the FastQ Screen Trimmomatic to remove the adaptor and low-quality sequences, and the filtered reads were mapped to GRCm38 with HISAT2 [127]. The expression level of each gene was quantified as FPKM https://toppgene.cchmc.org/ (fragments per kilobase of exon per million mapped fragments) and counts, and the DESeq2 algorithm ((http://cole-trapnell-lab.github.io/cufflinks/install/) was applied to filter the different expression genes (DEGs). The significance of the differentially expressed genes was identified based on the adjusted raw *p*-values to false discovery rate (FDR), <0.05, and fold changes (log2 FC > 1). Gene ontology (GO, http:\\www.geneontology.org\) and the Kyoto Encyclopedia of Genes and Genomes (KEGG, http:\\www.genome.jp\kegg\analyses) were explored to evaluate the biological function of DEGs [128]. 

### 4.7. Functional Annotation and Gene Network Analysis

Gene ontology (GO) enrichment analysis and DAVID annotation were used for functional annotation and pathway analysis, such as for the molecular function (MF), biological process (BP), and cellular component (CC), and the Kyoto Encyclopedia of Genes and Genomes (KEGG, http:\\www.genome.jp\kegg\analyses) was consulted to evaluate the biological functions of DEGs. GO terms with FDR (*q* < 0.05) were considered significantly enriched within the gene set. We performed protein−protein network analysis for all the DEGs using the STRING 10.5 database (https://string-db.org/), a useful tool for understanding metabolic pathways, and for predicting or developing genotype−phenotype associations.

### 4.8. Deconvolution of RNA-seq to Cell Type Composition

The mean expression levels of the signature genes from bulk RNA-seq were used as input for CIBERSORTx (http://cibersortx.stanford.edu) to calculate the relative distribution of the cell populations in vWAT derived from RA and IH conditions. For the CIBERSORTx cell-type reference we used information provided for leukocyte subtypes [129] and adipose tissues [25,50]. The deconvolution method generates absolute scores that can be interpreted as cell fractions for both inter- and intrasample comparisons. 

### 4.9. Quantitative Real-Time Polymerase Chain Reaction (qRT-PCR)

Total RNA was isolated from vWAT of RA and IH mice (*n* = 4/group) and qRT-PCR was performed using TaqMan gene expression. Briefly, total RNAs (1µg) were used to generate cDNA templates using High-Capacity cDNA Fast advance Master Mix, (# 4374966, Thermo Fisher Scientific, Waltham, MA, USA) according to the manufacturer’s instructions in 20 µL reaction as the following conditions: 25 °C for 10 min, 37 °C for 2 h, 85 °C for 5 min and the reaction could remain at 4 °C. The cDNA reaction contained the following: 2 µL of 10 RT Buffer, 0.8 µL of 25X dNTP Random Primers, 1.0 µL of MultiScribe Reverse Transcriptase, 1.0 µL of RNase inhibitor and 10 µL of nuclease-free water. Thirty-five ng of cDNA was used in each reaction in a total 25 µL reaction. Reaction conditions consisted of preincubation at 50 °C for 2 min and 95 °C for 10 min, followed by 40 cycles of 95 °C for 15 s and 60 °C for 1 min. The PCR reaction mix contained the following: 10 µL of TaqMan fast Advanced Master Mix (2×), 1.25 µL of TaqMan Assay primer (20×), 9.25 µL nuclease-free water. A series of genes were selected due to their roles in insulin resistance such as TBC1 domain family, member 4 (Tbc1d4) Mm00557659_m1); solute carrier family 27 (fatty acid transporter), member 1 (Slc27a1; Mm00449511_m1); phosphatidylinositol 3-kinase, catalytic, beta polypeptide (Pik3cb; Mm00659576_m1); phosphatidylinositol 3-kinase, regulatory subunit, polypeptide 1 (p85 alpha; Mm01282781_m1); insulin receptor substrate 1 (IRS1; Mm01278327_m1); cAMP responsive element binding protein 3-like 2 (Creb3l2; Mm00618366_m1); phosphoenolpyruvate carboxykinase 1, cytosolic (Pck1; Mm01247058_m1); glutamine fructose-6-phosphate transaminase 2 (Gfpt2); Mm00496565_m1); integrin beta 5 (Itgb5); Mm00439825_m1). qRT-PCR analysis was performed for these selected transcripts using QuantStudio 3 Real-Time PCR Systems (ThermoFisher Park, Berkeley, MO, USA). Three housekeeping genes (Thermo Fisher Scientific) were used including mouse *B*-actin (Mm00607939_s1), mouse 18s (Mm03928990_g1) and mouse TBP, TATA-Box Binding Protein, (Mm00446971_m1), and B-actin was used for normalization. We used *B*-actin in the normalization due to the very minor or no changes in eWAT between IH versus RA conditions compared to the other 2 housekeeping genes. Negative controls were used during cDNA reaction including no RNA and also no polymerase. All experiments were performed in triplicate. The quantification cycle (Cq) values were averaged and the difference between the housekeeping gene, Cq, and the gene of interest Cq was calculated to determine the relative expression of the gene of interest [62,118,130]. Several negative controls were used including: (a) no total RNA to monitor contamination and a primer-dimer formation that could produce false-positive results, (b) No reverse transcriptase control to monitor genomic DNA contamination, and (c) no amplification control to monitor background signal and probe stability. We also performed amplification efficiency of the genes by dilution of different concentrations of cDNA (1, 0.2, 0.04, 0.008, and 0.00016) from the same sample using the same qRT-PCR conditions (Appendix A).

### 4.10. Statistical Analysis

Data were expressed as means ± standard error (SE) and analyzed by *t*-tests. Two-group comparisons were made using two-tailed unpaired Student’s *t*-tests. Multiple comparisons were performed by one-way analysis of variance (ANOVA) following Tukey’s post hoc test (cite). For consistency in comparisons, significance in all figures were denoted as follows: * *p* < 0.05, ** *p* < 0.01, *** *p* < 0.001. Specific statistical analysis for each experiment is detailed in the corresponding figure legends.

## 5. Conclusions

Adipose tissue, an important endocrine organ, is involved in many cardiometabolic diseases including OSA-induced metabolic morbidities. Current snRNA-seq transcriptomic data revealed that vWAT heterogeneous cell population is altered by IH and the findings provide a framework for understanding the role of vWAT in metabolic disease in the context of OSA. Furthermore, deconvolution of bulk RNA-seq data should also contribute to our understanding of how IH influences cell-type heterogeneity and its impact in OSA and metabolic diseases. Our findings revealed a series of selected and integrative pathways that may possibly translate to identify therapeutic targets for patients with OSA.

## Figures and Tables

**Figure 1 ijms-22-00261-f001:**
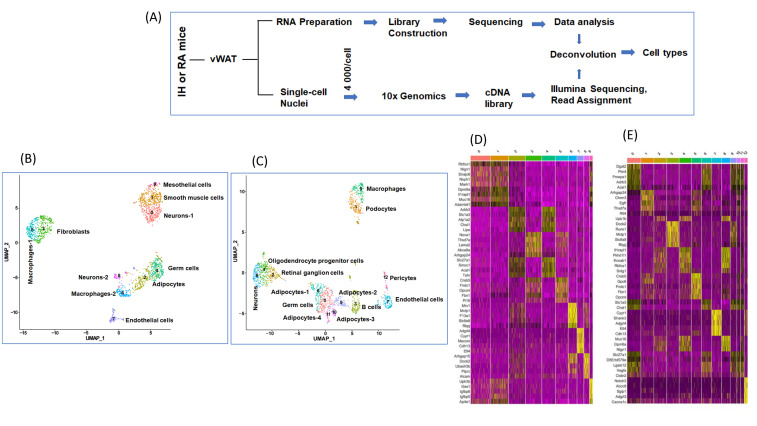
Single nuclei transcriptomic analysis (snRNA-seq) of murine epidydimal adipose tissues (vWAT) of mice exposed to IH or RA for 6 weeks. (**A**) Schema illustrates study design for snRNA-seq and bulk RNA-seq experiments. (**B**) Uniform manifold approximation and projection (UMAP) focuses on displaying the neighboring structure of the multidimensional manifold in few dimensions in vWAT for mice exposed to RA, *n* = 2. (**C**) UMAP in vWAT for mice exposed to IH, *n* = 2. Each dot corresponds to a single cell. (**D**) Heatmap for the 5 most enriched genes for each of the cell types in RA. (**E**) Heatmap for the 5 most enriched genes for each of the cell types in RA IH. Each cell type is represented by the top 5 genes ranked by false discovery rate (FDR) adjusted *p*-value of a Wilcoxon rank sum test between the average expression value for each cell type against the other average expression of the other cell types. Each column represents the average expression value for one cell type. Yellow color indicates the top 5 genes for each cell type.

**Figure 2 ijms-22-00261-f002:**
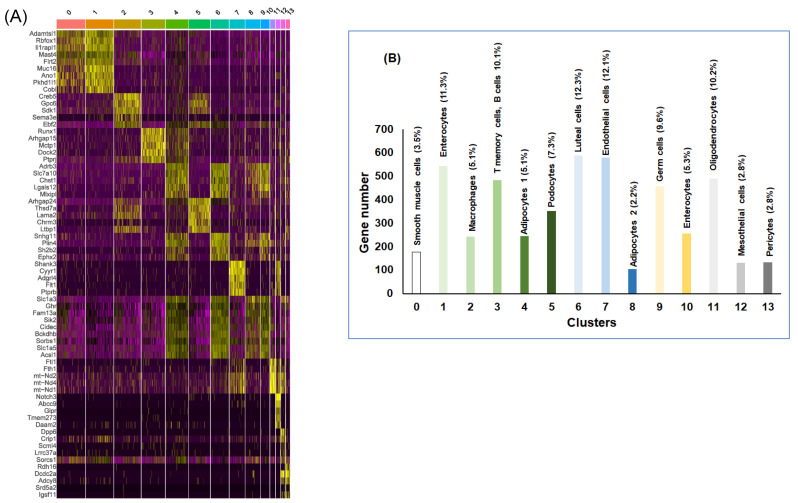
snRNA-seq for the statistically significant differentially expressed genes (DEGs) in vWAT of mice exposed to RA vs. IH for 6 weeks. (**A**) Heatmap of the 5 most differentially expressed genes (DEGs) of all 14 cell clusters. Each row of this heatmap is a gene determined to be a marker for one or more clusters, and each column is a cell. The cells are grouped by cluster. Each cell type is represented by the top 5 genes ranked by false discovery rate (FDR) adjusted *p*-value of a Wilcoxon rank sum test between the average expression per cluster value for each cell type (RA) against the other average subject expression of the other cell types for IH. Each column represents the average expression value for one cell type in RA vs. IH, *n* = 2/condition. (**B**) Histogram illustrates the gene number, percentage and cell types per cluster in DEGs in RA and IH. The heatmap shows the upregulated genes (ordered by decreasing *p*-value) in each cell type and selected enriched genes used for biological identification of each cluster (scale: Log_2_ fold change). Yellow color indicates the top 5 genes for each cell type.

**Figure 3 ijms-22-00261-f003:**
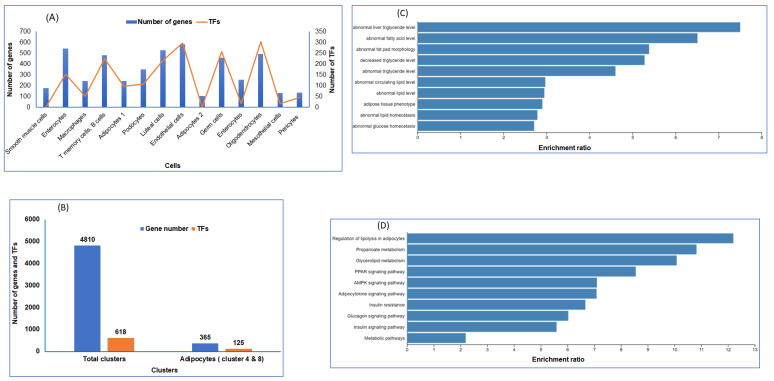
Transcription factors (TFs) of the differentially expressed genes in snRNA-seq. (**A**) TFs and gene numbers for each cell. (**B**) TFs and gene numbers for all cell clusters as well as cell types (clusters 4 and 8) which were identified as adipocytes. (**C**) Association of adipocyte DEGs and disease phenotypes in vWAT. (**D**) Association of adipocyte DEGs and metabolic KEGG pathways in vWAT.

**Figure 4 ijms-22-00261-f004:**
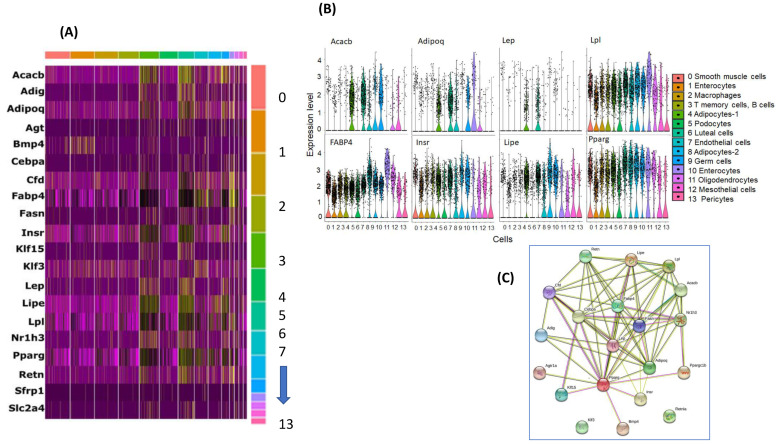
Selected adipocyte and adipogenesis genes in curated list reported in the literature. (**A**) Heatmap of the selected gene expression in each snRNA-seq cell type. (**B**) Violin plots for selected genes and their expression in snRNA-seq in each cell type, assigned by SingleR, using the expression profiles of individual cells. On the x axis, cluster number and y-axis the differentially expressed in RA vs. IH. (**C**) Gene networks of selected genes expressed in adipocytes and metabolic genes in vWAT of mice exposed to IH compared to RA.

**Figure 5 ijms-22-00261-f005:**
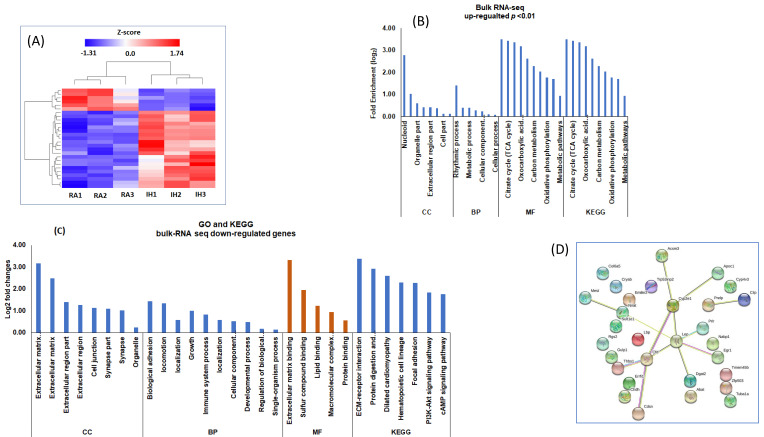
Bulk RNA-seq in vWAT of mice exposed to IH and RA for 6 weeks. The differentially expressed genes were used for gene ontology (GO), Kyoto Encyclopedia of Genes and Genomes (KEGG) pathways, and gene network interactions. (**A**) Hierarchical clustering analysis of gene expression profiles. Each column indicates a sample, whereas each row indicates a gene. Red color indicates upregulation and blue color downregulated genes. (**B**) Gene Ontology (GO) and KEGG for the upregulated differentially expressed bulk RNA-seq in vWAT from mice exposed to RA or IH for 6 weeks. (**C**) GO and KEGG for the downregulated differentially expressed genes. GO analysis was used to assess cellular components (CC), biological processes (BP), and molecular functions (MF). (**D**) Gene networks of the top 30 DEGs (15 upregulated and 15 downregulated) to construct gene-gene interactions using the STRING software. *n* = 3/condition.

**Figure 6 ijms-22-00261-f006:**
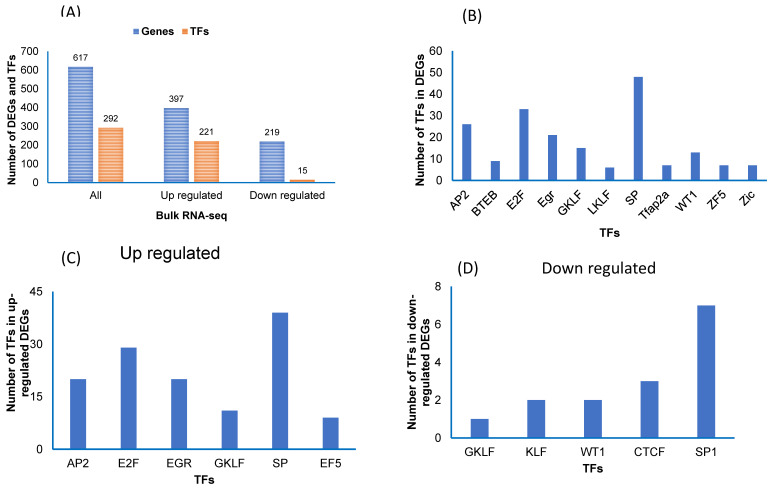
Transcription factors (TFs) in vWAT of the differentially expressed genes (DEGs) in mice exposed to RA and IH in individual cell types and in all cell types combined. (**A**) TFs for the DEGs for each cell type. (**B**) The major TFs identified for all cell types and adipocytes that were found in adipocyte cell types (clusters 4 and 8). (**C**) The major TFs were identified in the up regulated genes. (**D**) The major TFs were identified in the down regulated genes.

**Figure 7 ijms-22-00261-f007:**
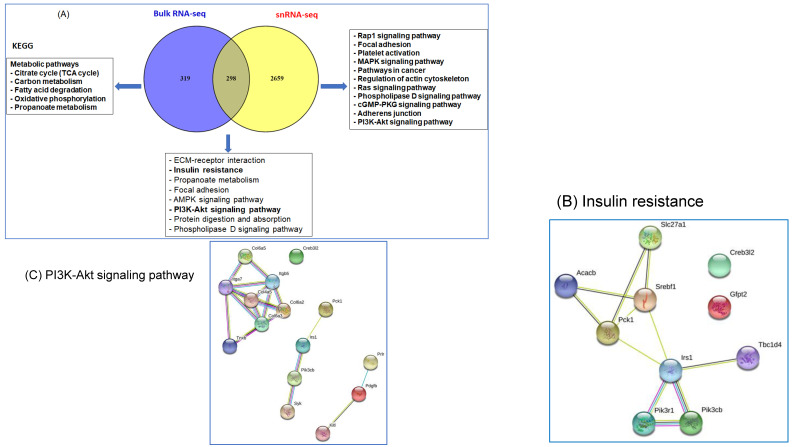
Comparison of the DEGs in bulk RNA-seq and snRNA-seq. (**A**) Venn diagram shows the comparison and overlap between the two transcriptomic analysis paradigms. The KEGG for 319 genes in bulk RNA-seq, 2659 genes in snRNA-seq and the overlap between the two systems 298 genes. (**B**) Gene networks for insulin resistance pathway. (**C**) Gene networks for PI3K-Akt signaling pathway. The gene−gene network analysis revealed that the DEGs are involved in various KEGG functional pathways, such as the AMPK signaling pathway, PPAR signaling pathway, and insulin resistance pathway.

**Figure 8 ijms-22-00261-f008:**
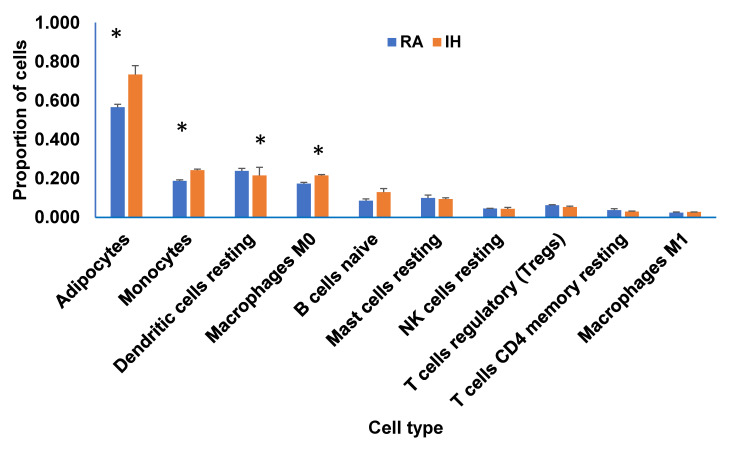
Deconvolution of bulk RNA-Seq of vWAT in mice exposed to IH compared to RA for 6 weeks. Bar graphs depict the frequency of each cell type in each condition from bulk RNA-seq data, as proportions of total cells inferred from markers of adipocytes; monocytes; dendritic cells resting; macrophages (M0); B cells; naïve; mast cells resting; NK cells resting; T cells regulatory (Tregs); T cells CD4; memory resting; and macrophages M1. *Y*-axis indicates the estimation of cell subtype proportion fraction of each cell type, *X*-axis indicates cell types. A comparison of decomposition estimates 547 genes from mouse immune system and 712 of mouse adipocytes as described in the Methods.

**Figure 9 ijms-22-00261-f009:**
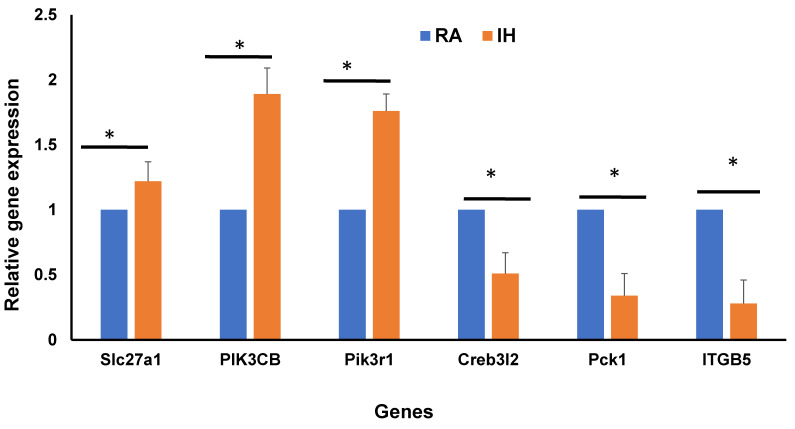
Validation of differentially expressed genes in bulk RNA-seq and snRNA-seq clusters involved in insulin resistance using qRT-PCR as shown in Venn diagram (Figure 7A). Total RNA was isolated from vWAT and analyzed by qRT-PCR analysis for 6 genes (3 upregulated and 3 downregulated). qPCR data were normalized to β-actin as an internal control. Data are presented as mean ± SD; *n* = 4/experimental condition; * indicates a significant difference at the *p* < 0.05 level. *n* = 4/condition.

**Table 1 ijms-22-00261-t001:** Cell types with their top 5 gene markers for vWAT in mice as separately determined for mice exposed to IH or RA.

Cluster	RA Cell Type	Gene Markers	IH Cell Type	Gene Markers
0	Neurons-1	Rbfox1, lgn1, Dnajc6, xph1, Mark1	Germ cells	Dgat2, Plin4, Pmepa1, Adrb3, Acsl1
1	Smooth muscle cells	Gpm6a, Il1rapl1, Muc16, Adamtsl1, Rbfox1	Podocytes	Arhgap24, Chrm3, Egfr Thsd7a, Rtl4
2	Adipocytes	Adrb3, Slc1a3, Atp1a2, Chst1, Lipe	Retinal ganglion cells	Upk1b, Dlgap1, Nkain2, Asxl3, Lars2
3	Fibroblasts	Nova1, Thsd7a, Lama2, Abca8a, Arhgap24	B cells	Dock2, Runx1, Mctp1, Slc9a9, Rbpj
4	Germ cells	Adrb3, Slc27a1, Smoc1, Acsl1, Tshr	Oligodendrocyte progenitor cells	Il1rapl1, Pkhd1l1, Kcnab1, Rbfox1, Sntg1
5	Macrophages-1	Creb5, Fndc1, Opcml, Fbn1, Pi16	Macrophages	Creb5, Gpc6, Fndc1, Fbn1, Opcml
6	Macrophages-2	Mrc1, Mctp1, F13a1, Slc9a9, Rbpj	Adipocytes-1	Adrb3, Plin4, Pmepa1, Slc1a3, Chst1
7	Endothelial cells	Adgrl4, Cyyr1, Mecom, Cdh13, Etl4	Endothelial cells	Cyyr1, Shank3, Adgrl4, Etl4, Cdh13
8	Neurons-2	Arhgap15, Dock2, Ubash3b, Ptprc, Alcam	Neurons	Muc16, Il1rapl1, Gpm6a, Nlgn1, Rbfox1
9	Mesothelial cells	Upk3b, Gas1, Igfbp6, Igfbp5, Ap4e1	Adipocytes-2	Mctp1, Slc1a3, Chst1, Adrb3, Slc27a1
10	NA		Adipocytes-3	Pmepa1, Chst1, Adrb3, Slc1a3, D5Ertd579e
11	NA		Adipocytes-4	Lgals12, Vegfa, Adrb3, Slc27a1, Clstn2
12	NA		Pericytes	Notch3, Abcc9, Sgip1, Adgrl3, Cacna1c

NA, not available.

**Table 2 ijms-22-00261-t002:** Cell types with their top 5 gene markers for vWAT in mice as determined from the combined data for mice exposed to IH or RA.

Cluster	Cell Type	Gene Markers
0	Smooth muscle cells	Adamtsl1, Rbfox1, Il1rapl1, Mast4, Flrt2
1	Enterocytes	Muc16, Ano1, Gpm6a, Pkhd1l1, Il1rapl1
2	Macrophages	Creb5, Fbn1, Gpc6, Fndc1, Sdk1, Opcml
3	T memory cells, B cells	Runx1, Arhgap15, Mctp1, Dock2, Slc9a9
4	Adipocytes-1	Vegfa, Adrb3, Slc7a10, Usp35, Slc1a3
5	Podocytes	Arhgap24, Thsd7a, Lama2, Abca8a, Chrm3
6	Luteal cells	Lgals12, Slc7a10, Snhg11, Dgat2, Plin4
7	Endothelial cells	Shank3, Cyyr1, Adgrl4, Flt1, Nova2
8	Adipocytes-2	Slc1a3, Adrb3, Ghr, Fam13a, Tenm4
9	Germ cells	Adrb3, Lgals12, Tshr, Bckdhb, Sorbs1
10	Enterocytes	Ftl1, Fth1, mt-Nd2, mt-Nd3, mt-Nd4
11	Oligodendrocytes	Notch3, Abcc9, Gipr, Tmem273, Aspn
12	Mesothelial cells	Dpp6, Cfap77, Dnah7c, Dnah7b, Crip1
13	Pericytes	Pkhd1, Rdh16, Dcdc2a, Adcy8, Ank3

## Data Availability

Data available on request.

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
