# Peer review of "Transcriptomic Changes of Murine Visceral Fat Exposed to Intermittent Hypoxia at Single Cell Resolution"

_ijms, 2020, doi:10.3390/ijms22010261_

Round 1
Reviewer 1 Report
Discussion is too long, authors can transfer them partially to introduction or reduce.
In discussion, they described "pathological changes" in sentences (line 402). It should be describe more detail.
The technique using single nucleus RNA sequencing (snRNA-seq) is good approach for understanding detailed molecular target genes (Good point). It is not well understood how intermittent hypoxia (IH) affect damage. They performed snRNA-seq and detected significant genes in WAT and focused on metabolic dysfunction. They found ECM is the top of KEGG pathway.
These findings provide new aspects for further IH mechanisms and ECM play important roles in metabolic dysfunctions.
It may be better they can show PCR data in selective genes than only snRNA-seq (weak point).
Author Response
We thank the reviewers for their insightful comments and critiques. We have incorporated their suggestions and revised the manuscript accordingly. We believe that their input has substantially improved our manuscript. Below is a point-by-point response, and changes are highlighted in red color in the revised redline version of the manuscript. We have also included a clean version as requested.
Comments and Suggestions for Authors
Reviewer 1
Q1: Discussion is too long, authors can transfer them partially to introduction or reduce.
Response 1: We thank the reviewer for the suggestion in modifying the discussion section. We edited the introduction and the discussion as requested.
Q2: In discussion, they described "pathological changes" in sentences (line 402). It should be described more detail.
Response 2: We edited the sentences.
Hypoxia causes vasoconstriction in the systemic and pulmonary circulation due to sympathetic nervous system activation, and recurrent hypoxia triggers systemic inflammation; alteration in transmural, intrathoracic, and cardiac pressure (Khalyfa et al., 2019; Salman et al., 2020). Furthermore, upper airway anatomy and collapsibility remain a fundamentally important pathophysiological factor. However non‐anatomical factors, such as impaired muscle responsiveness, low arousal threshold, high loop gain, rostral fluid shifts, lung volume, additionally play a variable role (Rana et al., 2020; Chen et al., 2020).
- Khalyfa A, Castro-Grattoni AL, Gozal D. Cardiovascular morbidities of obstructive sleep apnea and the role of circulating extracellular vesicles. Ther Adv Respir Dis. 2019 Jan-Dec 2019; Dec;13:1753466619895229. doi: 10.1177/1753466619895229.
- Salman LA, Shulman R, Cohen JB. Obstructive Sleep Apnea, Hypertension, and Cardiovascular Risk: Epidemiology, Pathophysiology, and Management. Curr Cardiol Rep. 2020 Jan 18;22(2):6. doi: 10.1007/s11886-020-1257-y.
- Rana D, Torrilus C, Ahmad W, Okam NA, Fatima T, Jahan N Obstructive Sleep Apnea and Cardiovascular Morbidities: Cureus. 2020 Sep 13;12(9):e10424. doi: 10.7759/cureus.10424.
- Chen LD, Chen Q, Lin XJ, Chen QS, Huang YZ, Wu RH, Lin GF, Huang XY, Lin QC. Effect of chronic intermittent hypoxia on gene expression profiles of rat liver: a better understanding of OSA-related liver disease. Sleep Breath. 2020 Jun;24(2):761-770. doi: 10.1007/s11325-019-01987-0. Epub 2019 Dec 16.
The technique using single nucleus RNA sequencing (snRNA-seq) is a good approach for understanding detailed molecular target genes (Good point). It is not well understood how intermittent hypoxia (IH) promotes tissue or organ dysfunction. The authors performed snRNA-seq and detected significant changes in multiple genes in WAT and focused on metabolic dysfunction. They found ECM is the top of the KEGG pathway. These findings provide new aspects for further IH mechanisms and ECM plays important roles in metabolic dysfunctions.
We thank the reviewer for this excellent point about IH, cell type, and ECM pathway. We believe the ECM pathway is very important for understating the effect of IH and metabolic dysfunction.
Q3: It may be better they can show PCR data in selective genes than only snRNA-seq (weak point).
Response 3: We appreciate the reviewer excellent remark. Unfortunately, we do not have any additional adipose tissues to perform this experiment, we will consider this in our future experiments.
Reviewer 2 Report
The authors have used snRNA-seq and aggregate RNA-seq on non-obese mouse vWAT to describe heterogeneity of the cellular response to CIH, mimicking OSA. This is an exploratory study to provide insight into the metabolic morbidity of OSA, as vWAT is often overlooked as a major endocrine organ. The authors provide insight into pathways and processes that are altered due to CIH, at a global and cellular level.
The manuscript would benefit from a conclusion paragraph which focuses on the MAJOR differences between gene programs in RA and CIH. Please describe specifically which important pathways are up or down regulated compared to RA and how these pathways might contribute to the clinical metabolic OSA manifestations.
(1) The way the figures are currently displayed, it is impossible to read the very small text that correspond. Even online, when enlarging the figure, the text is blurry. This reviewer suggests either enlarging the figures or retyping the text within figures with a larger font.
(2) There are grammatical and/or formatting errors in the text that make reading the text ambiguous, for example line 69, table 1, line 166, line 200, 339. And other spelling and grammatical errors through the text.
(3) Be sure to define all abbreviations, ie. line 255, 256, 272
(4) The authors should be aware that when using Transcription Factor analysis tools, the transcription factors that may be responsible for the input DEGs are merely suggestions based on an algorithm. The way the text describes these data is that these TF's are actually more abundant in the gene set, which I don't believe is true. Please be sure to describe these data so the reader understands clearly about the TFs, whether they are predicted, or whether you just pulled out the TFs from your list of DEGs.
(5) Is there significance in Figure 9? No asterisks are indicated.
(6) Please state how many animals were sampled. The methods only indicate 4,000 cells per sample.
(7) Where was the fat taken from. As described in the Discussion, depending on the depot, cell composition of adipose tissue can vary widely. Please be specific about where in the animal the visceral fat was obtained.
Author Response
We thank the reviewer for their insightful comments and critiques. We have incorporated their suggestions and revised the manuscript accordingly. We believe that their input has substantially improved our manuscript. Below is a point-by-point response, and changes are highlighted in red color in the revised redline version of the manuscript. We have also included a clean version as requested.
Reviewer 2
The authors have used snRNA-seq and aggregate RNA-seq on non-obese mouse vWAT to describe heterogeneity of the cellular response to CIH, mimicking OSA. This is an exploratory study to provide insight into the metabolic morbidity of OSA, as vWAT is often overlooked as a major endocrine organ. The authors provide insight into pathways and processes that are altered due to CIH, at a global and cellular level.
We thank the reviewer for a succinct and insightful summary of our findings.
The manuscript would benefit from a conclusion paragraph which focuses on the MAJOR differences between gene programs in RA and CIH. Please describe specifically which important pathways are up or down regulated compared to RA and how these pathways might contribute to the clinical metabolic OSA manifestations.
Response 1: We thank the reviewer for raising this very important point. We have indicated which pathways are altered in IH compared to RA, however, we didn’t elaborate on how these pathways might contribute to the clinical metabolic OSA manifestations. Now, we edited this part of the Discussion. We identified 91 KEGG pathways in snRNA-seq in all cells combined, while we identified 21 pathways in bulk RNA-seq. Among those pathways, the PI3K-Akt signaling pathway, AMPK signaling pathway, and insulin resistance pathway were shared in both snRNA-seq and bulk-RNA-seq, as well as their associated genes (Supplementary Tables 3 and 4). Those pathways are involved in OSA with metabolic dysfunction
- Huang X, Liu G, Guo J, Su Z. The PI3K/AKT pathway in obesity and type 2 diabetes. Int J Biol Sci. 2018 Aug 6;14(11):1483-1496. doi: 10.7150/ijbs.27173. eCollection 2018. PMID: 30263000.
- Yang CH, Shen YJ, Lai CJ, Kou YR. Inflammatory Role of ROS-Sensitive AMP-Activated Protein Kinase in the Hypersensitivity of Lung Vagal C Fibers Induced by Intermittent Hypoxia in Rats. Front Physiol. 2016 Jun 27;7:263. doi: 10.3389/fphys.2016.00263. eCollection 2016. PMID: 27445853.
- Evans AM, Mahmoud AD, Moral-Sanz J, Hartmann S. The emerging role of AMPK in the regulation of breathing and oxygen supply. Biochem J. 2016 Sep 1;473(17):2561-72. doi: 10.1042/BCJ20160002. PMID: 2757402
- Xu H, Liang C, Zou J, Yi H, Guan J, Gu M, Feng Y, Yin S. Interaction between obstructive sleep apnea and short sleep duration on insulin resistance: a large-scale study: OSA, short sleep duration and insulin resistance. Respir Res. 2020 Jun 16;21(1):151. doi: 10.1186/s12931-020-01416-x. PMID: 32546151.
- The way the figures are currently displayed, it is impossible to read the very small text that correspond. Even online, when enlarging the figure, the text is blurry. This reviewer suggests either enlarging the figures or retyping the text within figures with a larger font.
Response 1: We thank the reviewer for improving figures resolution. Since many panels are included in each figure, it would be difficult to make each figure perfect. We will re-edit the figure for the final version.
- There are grammatical and/or formatting errors in the text that make reading the text ambiguous, for example line 69, table 1, line 166, line 200, 339. And other spelling and grammatical errors through the text.
Response 2: We thank the reviewer and made the appropriate correction in the text, figures and tables.
- Be sure to define all abbreviations, ie. line 255, 256, 272
Response 3: We included the abbreviations as suggested.
- The authors should be aware that when using Transcription Factor analysis tools, the transcription factors that may be responsible for the input DEGs are merely suggestions based on an algorithm. The way the text describes these data is that these TF's are actually more abundant in the gene set, which I don't believe is true. Please be sure to describe these data so the reader understands clearly about the TFs, whether they are predicted, or whether you just pulled out the TFs from your list of DEGs.
Response 4: We thank the reviewer for this very good point. The purpose of the study is to identify which TFs are involved in the statistically significant DEGs in IH versus RA. In this study, we are not discussing how these TFs are regulated or controlling other genes via epigenetic mechanisms, instead our goal was to identify those TFs in each cell type. Transcription factors are essential to cellular function and are vital molecules in the control of gene expression. They bind to specific sequences of DNA and control the transcription of DNA into mRNA. The gene expression is regulated through either activation or repression of transcription factors, which are essential for a range of key cellular processes.
- Is there significance in Figure 9? No asterisks are indicated.
Response: We added as suggested. We inserted a new figure with asterisks.
- Please state how many animals were sampled. The methods only indicate 4,000 cells per sample.
Response: We included in the Methods and Figure legend, n=2 for snRNA-seq and n=4 for bulk RNA-seq..
- Where was the fat taken from. As described in the Discussion, depending on the depot, cell composition of adipose tissue can vary widely. Please be specific about where in the animal the visceral fat was obtained.
Response: We used epididymal white adipose tissue (eWAT) and this is indicated in the methods.